# The Impact of Rye and Barley Malt and Different Strains of *Saccharomyces cerevisiae* on Beer Volatilome

Noemi Tocci [1], Gian Marco Riccio [1], Abirami Ramu Ganesan [1,2], Philipp Hoellrigl [1], Peter Robatscher [1] and Lorenza Conterno [1,*]

1   Laimburg Research Centre, Laimburg 6, 39051 Pfatten/Vadena, BZ, Italy; noemi.tocci@laimburg.it (N.T.); gianmarco.riccio@laimburg.it (G.M.R.); abirami.ganesan@nibio.no (A.R.G.); philipp.hoellrigl@hotmail.de (P.H.); peter.robatscher@laimburg.it (P.R.)
2   Division of Food Production and Society, Biomarine Resource Valorisation, Norwegian Institute of Bioeconomy Research, Kudalsveien 6, NO-8027 Bodø, Norway
*   Correspondence: lorenza.conterno@laimburg.it

**Abstract:** Craft breweries are continuously searching for beers made with locally produced raw materials and unique flavor profiles to respond to consumer requests. We explored the behavior of three commercial strains of *Saccharomyces cerevisiae* in the fermentation of ale beer with a high prevalence of rye malt in comparison to pure barley malt. In total, 34 volatile organic compounds were identified, with esters and alcohols being the quantitatively most abundant classes. The yeast strain appeared to impart more differences in the beer's volatile profile compared to malt. In particular, *S. cerevisiae* var. *diastaticus* Y2 strain was associated with a higher production of esters, while strain *S. cerevisiae* Y3 was correlated to the higher amounts of terpenes together with the lowest relative abundance of volatile acids. Our findings encourage further investigation of the fermentation performance of several yeast strains to produce beers with unique flavors.

**Keywords:** craft beer; rye malt; volatile organic compounds; mass spectrometry; *S. cerevisiae* var. *diastaticus*





## 1. Introduction

In recent decades, the emergence of small and independent breweries has deeply transformed the global beer market, ending a century of domination by a few global multinational macro-breweries and their uniform beer style [1]. According to the Brewers of Europe, in 2021, the European total beer production was about 34.2 billion liters [2] from 9436 breweries. France, the UK, and Switzerland, followed by Germany and Italy, represent the top five producers of craft beer in Europe, with constant growth and a compound annual growth rate (CAGR) of 7.87% from 2022 to 2027 [3]. In Italy, starting in 2000, the number of craft breweries grew from 60 to more than 800, as recorded in 2021 [4]. The small breweries respond to the request of consumers searching for beers characterized by unique profiles, made with local raw materials, and produced by master brewers with outstanding beer culture [5]. Beer organoleptic traits are deeply dependent on raw ingredients and on the beer production process. Consequently, to develop a peculiar beer, the choice of specific ingredients, yeast strains, and the development of an ad hoc production process are of prior importance for its market positioning.

The main actors in beer are water, malt, hops, and yeasts. Barley, in its malted form, is the most used cereal. The contribution of barley to beer flavor is largely influenced by the malting process, as well as by its chemical composition [6]. Different kinds of barley malt are commercially available, and the combination with specific hops and yeasts drives the production of a huge variety of beers. The introduction of locally grown cereals or pseudocereals in different proportions generates further diversification. Among cereals, rye (*Secale cereale* L.) is a crop that is uncommonly tolerant to cool climates and less fertile soil conditions, often growing where wheat cannot be grown. It has a long tradition of uses in

the production of alcoholic beverages such as the kvass, a Russian fermented beverage, the Finnish sahati, and the traditional Bavarian Roggenbier [7]. In modern brewing, rye enters the grist of some beer styles like Belgian Pale Ale [8], Saison, and Specialty India Pale Ale (Rye IPA), conferring a spicy and pungent aroma to complete the barley taste [9]. Despite a renewed interest in the application of rye grain and malt in brewing being registered in the past decade [10], because of the several challenges to brewers, it is often only moderately used in beer making (10–30%). Limited information is available on the malting of rye and brewing performance of rye varieties. Therefore, scientific investigation on brewing with rye malt is actually a very active sector of interest [10,11].

Another important source of flavor diversification in beer is represented by yeast. The selection of yeast, in fact, determines the final aroma and volatile active compounds of beer, contributing to the production of alcohols and esters detectable in the final product [12,13]. Two major yeast species are currently used in breweries: the *Saccharomyces cerevisiae*, mainly responsible for ale fermentation, and the *Saccharomyces pastorianus*, a hybrid species responsible for lager fermentation [14]. Among *S. cereviasie*, the strains defined as belonging to the variety *diastaticus* exhibit extracellular glucoamylase activity, a distinctive phenotype encoded by STA genes, allowing the degradation of starch and dextrins into glucose units from non-reducing ends. This strain is often associated with spoilage, which may occur during both primary and secondary fermentation [15], leading to the production of a higher amount of ethanol and off-flavors. On the other hand, its super-attenuating ability may be considered interesting and desirable in the production of certain beers, namely low-carbohydrate beers or those produced by high-gravity brewing [16]. Being characterized by diverse genetic variation, the choice of a peculiar strain of *S. cerevisiae* var. *diastaticus* may be helpful in developing new flavor profiles suitable for a specific brewing context [17]. Diastatic strains are often associated with Saison-style beers, which have been growing in popularity for some years [18].

With the aim of exploring the application of other cereals and uncommon yeasts in brewing, the present study investigates the effect of novel malt composition, including a malt mix made in prevalence with rye, in combination with yeast characterized by different genetic features, on the physicochemical parameters and volatile compounds of beers. Moreover, a sensory evaluation of the new products was performed.

## 2. Materials and Methods
### 2.1. Materials

The following were used in the beer-making process: Pilsner malt, Munich malt, rye malt, rye caramel malt, and barley caramel malt (Weyermann, Bamberg, Germany); Bravo, Hüll melon, Saphir, and England Target hop; pellets (Hopsteiner, Mainburg, Germany); and active dry yeast (Lallemand brewing, Montreal, CQ, Canada). All analytical reagents and kits for the enzymatic analysis were purchased from CDR s.r.l. (Ginestra Fiorentina, Italy) and from Exacta Optec (Verona, Italy).

#### Experimental Setup

This study focused on barley (100%) (BB) and rye–barley (55–45%) (RB) beer production using different yeast strains. Rye is a crop characteristic of mountain areas, where this study was carried out, and is uncommonly tolerant to cool climates and less fertile soil conditions, often growing where wheat cannot be grown. In order to develop a local product and overcome technical problems arising from the high amount of arabinoxylan in rye [10,11], the rye–barley grist recipe was formulated with 55% of rye malt together with 45% of barley malt. All beers were Amber-style ale. Three different yeast strains, *Saccharomyces cerevisiae* London ESB (Y1), *S. cerevisiae* var. *diastaticus* Belle Saison (Y2), and *S. cerevisiae* BRY-97 West Coast Ale (Y3) as dry active yeast, have been inoculated in purity. The yeast strains were chosen for their brewing properties (as described by the manufacturer), Y1 does not utilize maltotriose, and it is referred to as a moderate ester producer, allowing malt aromas and flavors to prevail. Y2 shows high attenuation power,

can utilize some types of dextrins, and is capable of producing citrus and pepper aroma notes. Y3 is a neutral strain that can promote hop biotransformation and emphasize hop flavor and aroma. Y3 was selected by the manufacturer for beer made with rye malt, and, in this study, it was used only for RB beer production as a parameter of comparison to study the influence of yeasts not conventionally used for rye malt fermentation.

### *2.2. Beer Production*

#### 2.2.1. Milling

Malt milling was carried out with a malt mill (Kompakt MattMill, Kirker, Germany), producing grain particles with a size of about 1.3 mm.

#### 2.2.2. Mashing

Mashing was performed in a 10 L Braumeisterbrewer (Speidel Braumeister, Ofterdinger, Germany) according to the process detailed in Table 1, which is different for the two BB and RB grists (Table 2). Briefly, BB mashing begins at 60 °C, followed by a 63 °C and a 73 °C step carried out for 25 and 40 min, respectively, and by a final 78 °C step lasting for 10 min.; RB mashing begins at 35 °C, followed by 45 °C, 55 °C, 63 °C, and 73 °C steps carried out for 15, 25, 25, and 40 min, respectively. A final step at 78 °C lasted for 10 min. The two mashing programs were set after preliminary tests. Lautering and sparging with 1 L of warm tap water were performed, separating the solid from the wort.

**Table 1.** Mashing profiles used in this study. BB: barley malt beer; RB: rye–barley malt beer.

| Profile | Mashing | β-Glucanase Activity | Protein Rest | β-Amylase Rest | α-Amylase Rest | Lautering | Boiling |
|---------|---------|----------------------|--------------|----------------|----------------|-----------|---------|
| BB | 60 °C | 63 °C 25 min | none | 73 °C 40 min | none | 78 °C 10 min | 100 °C 80 min |
| RB | 35 °C | 45 °C 15 min | 55 °C 25 min | 63 °C 25 min | 73 °C 40 min | 78 °C 10 min | 100 °C 80 min |

**Table 2.** Ingredients used for the five different beer recipes (n = 3). Amount: if not differently specified, it is the quantity used for each 10 L batch. BB: barley malt beer; RB: rye–barley malt beer.

| Recipe | Ingredient | Amount |
|--------|-----------|--------|
| BB1, BB2 | Pilsner malt | 1000 g |
| | Munich malt | 1200 g |
| | Barley caramel malt | 300 g |
| | Hop pellet | 13.6 g |
| RB1, RB2, RB3 | Pilsner malt | 1125 g |
| | Rye malt | 1250 g |
| | Rye caramel malt | 125 g |
| | Hop pellet | 13.6 g |
| **Recipe** | **Yeast Strain** | **Amount** |
| BB1 | Active dry yeast Y1 | 0.5 (g/L) |
| BB2 | Active dry yeast Y2 | 0.5 (g/L) |
| RB1 | Active dry yeast Y1 | 0.5 (g/L) |
| RB2 | Active dry yeast Y2 | 0.5 (g/L) |
| RB3 | Active dry yeast Y3 | 0.5 (g/L) |

#### 2.2.3. Wort Boiling

After removing the spent grains, the wort was heated to reach the boiling point set at 100 °C. At this stage, hop pellets (40 g) were added to release bitterness. Subsequently, 10 g of hop pellet was added for the final 10 min of boiling. Hop mix (Bravo–Hüll melon–Saphir–England Target: 1.1–3.4–7.1–2.0 g/10 L, respectively) was the same for all the beers and was defined after preliminary tests. A manual whirlpool was applied to clarify the

wort, and the mixture was cooled down to 20 °C with the aid of tap water circulating into a cooling coil tool.

The density was adjusted to the desired 12.5 °Plato (12.5 °Brix; 1.048 SG) achieved by diluting the must with water according to Formula (1):

$$water\ addition\ [\text{L}] = deflector\ quantity/volume[\text{L}] \ \times \left(\frac{Actual - stock\ wort}{Target - stock\ wort} - 1\right) \quad (1)$$

### 2.2.4. Fermentation

The wort was transferred into the fermentation tank through an aseptic funnel. At this stage, yeast (>2.5 billion cells/mL) was added at a dose of 0.5 g/L after rehydration performed according to the manufacturer's instructions. Each tank was equipped with an airlock valve. At the end of the fermentation, carried out at room temperature (20 °C) and monitored by measuring the $CO_2$ loss, the beers were bottled. At bottling, 8 g of sugar was added for second fermentation in every 750 mL bottle. One bottle for each batch was equipped with a manometer to monitor the pressure increase. The other bottles were closed using crown cork and incubated at 20 °C for one week, occurring prior to maturation at 10 °C for 3 to 4 months. Each beer variant (Table 2) was produced in triplicate.

### 2.3. Physicochemical Parameters

Physicochemical parameters were determined for wort and at the end of the first and second fermentations. The $CO_2$ release was measured by recording the weight loss during the first fermentation process. The total soluble solid content (°Brix was determined with a digital pocket refractometer (PAL-BX/RI, Atago, Bellevue, WA, USA) and converted in °Plato. The pH was measured with the aid of the XS digital pH Meter (XS Instruments, Carpi, Italy). The fermentable sugars (maltose, glucose, and fructose) and the alcohol content (ethanol) were measured following the international methods through the CDR BeerLab® Touch (CDR s.r.l., Ginestra Fiorentina, Italy), according to the manufacturer's instructions. The method applied for the measurement of glucose and fructose does not allow for the quantification of the two compounds separately; therefore, in the manuscript, they will be indicated together as Glu-Fru. Free amino nitrogen, ammonium ($NH_4^+$), and acetic acid were measured enzymatically using a semiautomatic analyzer Exacta Italo S® (ITALO S, Exacta Optech Labcenter, San Prospero, Italy), following the manufacturer's instructions.

### 2.4. Volatile Component Analysis

A GC-MS (QP 2010 SE, Shimadzu, Milan, Italy) equipped with an autosampler (AOC-600, Shimadzu, Milan, Italy) was used to determine the volatile organic compounds (VOCs). VOCs were analyzed according to the method described by Ravasio et al. [19] with slight modifications. All samples were incubated for 10 min at 40 °C under shaking (250 rpm), and then the VOCs in the headspace were collected on divinylbenzene/carboxen/polydimethylsiloxane fiber (DVB-CAR-PDMS) coating 50/30 μm (50 μm for the DVB layer and 30 μm for the CAR/PDMS layer) (Supelco, Milan, Italy) for 40 min at 40 °C under shaking (250 rpm); the fiber was introduced into the injector (set at 240 °C) for 5 min with a split ratio of 17. The DVB-CAR-PDMS fiber has been used because it is the most appropriate for capturing a broad range of VOCs belonging to different chemical classes in beer products and, therefore, suitable for the analysis of a whole profile of VOCs. DVB (divinylbenzene) and CAR (Carboxen) are porous particles for absorbing the VOCs, and PDMS (polydimethylsiloxane) is a polymeric film used for absorbing VOCs and as binder for DVB and CAR. The combination of these three materials on a fiber permits the highest efficiency in capturing volatiles and semi-volatiles (C3–C20) [20].

Separation was performed on a 60 m × 0.25 mm × 0.25 mm ZB-WAX fused-silica capillary column (Phenomenex, Milan, Italy), and helium was used as carrier gas with a flow rate of 1.4 mL/min. The GC oven temperature was set to 40 °C for 4 min, and then the temperature was increased to 250 °C at a rate of 6 °C/min and held for 5 min. An

electron ionization (EI) at 70 eV was applied, the transfer line temperature was 250 °C, and the mass spectrometry detector temperature was set at 200 °C. Mass spectra were acquired in full scan mode from 35 to 350 m/z. After each run, the SPME fiber was cleaned at 270 °C for 15 min. The VOCs were identified by comparing the acquired spectra to a mass spectral database (NIST 2017 GC-MS library); the relative concentration of each volatile was expressed as mg/L of 1-heptanol (internal standard—IS) by comparing the peak area of each analyte with the peak area of the IS. A quality control chromatogram can be seen in Figure S1 (Supplementary Materials).

### 2.5. Sensory Analysis

The sensory evaluation of beer was carried out by a panel of 37 semi-trained judges with prior knowledge about the sensory characteristics of beer. All the panelists evaluated five types of beer in distinctive sessions. The beer samples were served in an alphanumeric sequence as a blinded sample. Fifty milliliters of beer samples were supplied to the panelist. A nine-point hedonic scale was used to define five attributes, including aroma, acidity, perlage, bitterness, and overall impression: 9: like extremely; 8: like very much; 7: like moderately; 6: like slightly; 5: neither like nor dislike; 4: dislike slightly; 3: dislike moderately; 2: dislike very much; 1: dislike extremely. Off-flavor presence/absence question was also asked of the panelist.

### 2.6. Statistical Methods

For statistical analyses, three replicates per sample were considered. The data sets were analyzed for significant differences between the groups by one-way analysis of variance (ANOVA) followed by Tuckey's post hoc test, and $p < 0.05$ was accepted as statistically significant. The influence of the variables "yeast" and "malt" on beer was investigated through two-way ANOVA followed by Bonferroni's correction. The chemical data were compared for similarities using principal component analysis (PCA). MetaboAnalyst 5.0 [21] was used to perform all the above-mentioned analyses. The sensory data were evaluated by Friedman Test with n = 37 test subjects.

## 3. Results and Discussion

The main quality characteristics of beer are appearance, aroma, flavor, and mouthfeel, which result from the fermentation process and mainly depend on the raw materials utilized and on the yeast performance [22]. In this study, a 100% barley grist and a rye–barley grist (55–45%), together with three different yeast strains, were evaluated for the fermentation parameters and the beer aroma described by the measured VOCs and by the sensory evaluation.

### 3.1. Fermentation Monitoring—$CO_2$ Production

As expected, during the first fermentation, the amount of $CO_2$ released increased, reaching the highest values 2, 3, and 7 days after yeast inoculum for beers fermented with Y1 (613.92 ± 2.43 mmol/L), Y3 (716.51 ± 23.93 mmol/L), and Y2 (905.69 ± 71.15 mmol/L), respectively. The results, as shown in Figure 1, indicate that the yeast induces statistically significant differences ($p < 0.05$) among the samples. The beers produced with *S. cerevisiae* strain Y1 and Y3 released a lower $CO_2$ amount in comparison with the release recorded for *S. cerevisiae* var. *diastaticus* (Y2); this can be explained by its ability to synthesize extracellular glucoamylase, degrading starches and dextrins to fermentable sugars, resulting in a higher production of alcohol and $CO_2$ in comparison with beers fermented with the other *S. cerevisiae* strains unable to metabolize starch and dextrins.

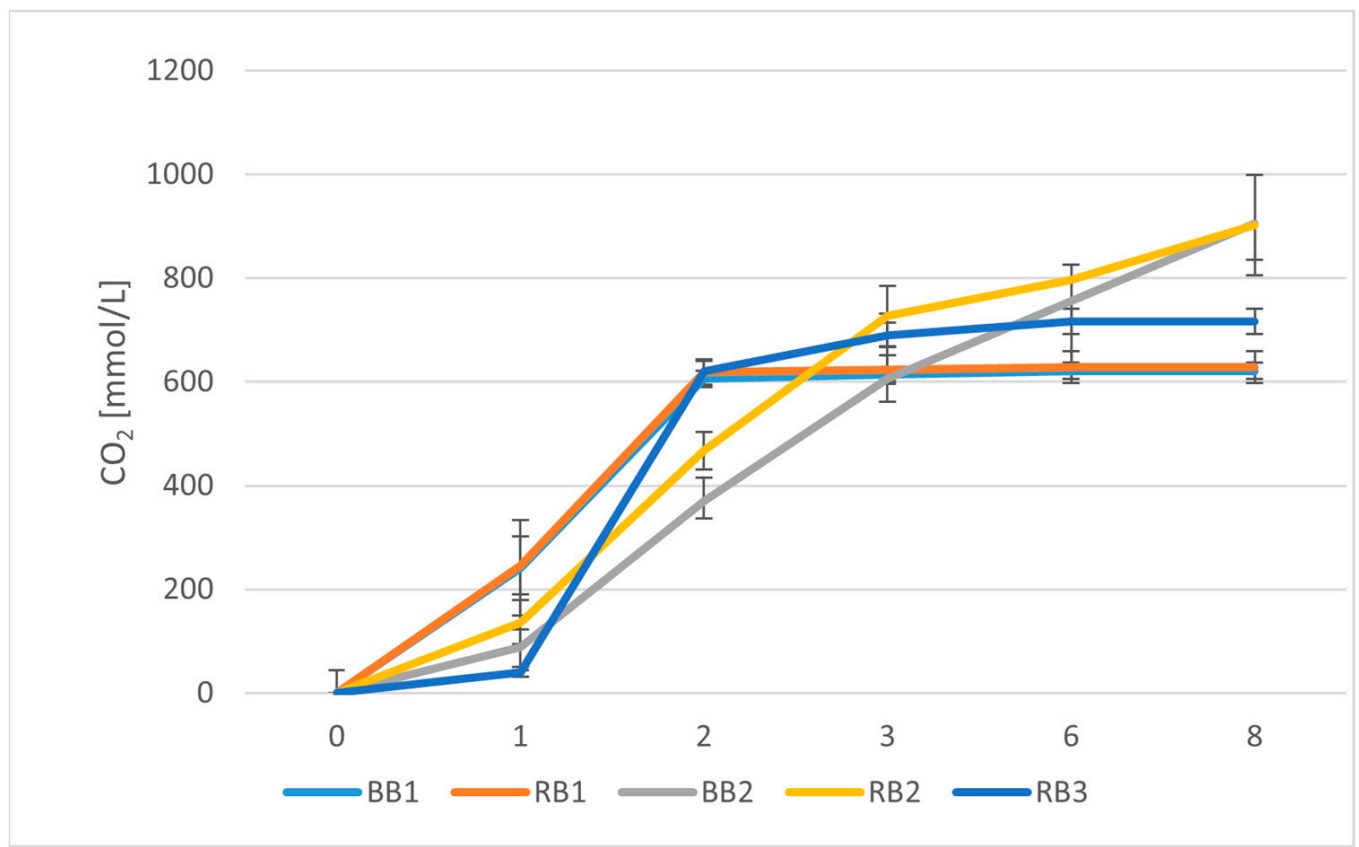

**Figure 1.** $CO_2$ production during fermentation by different *S. cerevisiae* yeast strains fermenting beer produced starting from 100% barley and rye–barley (55–45%) malt. BB1 = barely malt + *S. cerevisiae* strain Y1; RB1 = rye–barley malt + *S. cerevisiae* strain Y1; BB2 = barely malt + diastatic *S. cerevisiae* strain Y2; RB2 = rye–barley malt + diastatic *S. cerevisiae* strain Y2; RB3 = rye–barley malt + *S. cerevisiae* strain Y3. Each data point represents the average of three replicates, with error bars indicating the standard deviation.

### 3.2. Physicochemical Parameters

Physicochemical parameters were analyzed in wort and in beers at the end of the first and second fermentation. Features measured on products at a pre- and post-fermentation stage are reported in Table 3, together with ANOVA analysis and Tukey's multiple comparison post hoc test. In wort, significative differences were highlighted for maltose, glucose, fructose, and ammonium, while at the end of both fermentations, the samples showed significant differences for four variables (pH, alcohol, ammonium, and acetic acid after the first fermentation, and pH, alcohol, maltose, and ammonium at the end of the bottle fermentation and maturation).

**Table 3.** Physicochemical analysis of beers produced with different *S. cerevisiae* yeast strains starting from worts produced with 100% barley or rye–barley (55–45%) malt.

| (a) Wort | BB1 | RB1 | BB2 | RB2 | RB3 |
|---|---|---|---|---|---|
| pH | 5.96 ± 0.04 | 6.17 ± 0.46 | 5.89 ± 0.08 | 5.89 ± 0.05 | 5.93 ± 0.05 |
| Plato | 12.6 ± 0.01 | 12.6 ± 0.01 | 12.47 ± 0.12 | 12.27 ± 0.58 | 12.43 ± 0.40 |
| Maltose (g/L) | 68.00 ± 1.00 [a] | 68.33 ± 4.16 [ab] | 65.00 ± 7.00 [ab] | 58.00 ± 15.39 [ab] | 64.11 ± 3.39 [b] |
| Glu-Fru (g/L) | 33.48 ± 3.42 [a] | 12.76 ± 15.23 [a] | 6.91 ± 0.61 [d] | 9.30 ± 0.60 [c] | 12.14 ± 0.44 [b] |
| FAN (mg/L) | 246.00 ± 11.53 | 173.67 ± 20.50 | 230.33 ± 18.01 | 227.00 ± 8.72 | 230.33 ± 2.52 |
| $NH_4^+$ (mg/L) | 67.33 ± 5.03 [c] | 81.50 ± 5.50 [b] | 60.33 ± 7.02 [c] | 68.00 ± 7.21 [c] | 97.00 ± 4.36 [a] |

**Table 3.** *Cont.*

| (b) first fermentation | BB1 | RB1 | BB2 | RB2 | RB3 |
|---|---|---|---|---|---|
| pH | 4.60 ± 0.02 [b] | 4.71 ± 0.01 [a] | 4.28 ± 0.0 [c] | 4.29 ± 0.04 [c] | 4.33 ± 0.04 [b] |
| Alcohol (% *v/v*) | 3.57 ± 0.09 [b] | 3.67 ± 0.14 [b] | 5.23 ± 0.40 [a] | 5.20 ± 0.63 [a] | 4.37 ± 0.31 [a] |
| Maltose (g/L) | 5.70 ± 9.44 | 2.90 ± 4.95 | 5.07 ± 8.60 | 5.06 ± 8.60 | 0.2 ± 0.17 |
| Glu-Fru (g/L) | 0.01 ± 0.01 | 0.05 ± 0.02 | 0.89 ± 0.71 | 0.55 ± 0.79 | 0.01 ± 0.01 |
| FAN (mg/L) | 58.00 ± 14.14 | 57.00 ± 24.04 | 54.67 ± 16.07 | 49.00 ± 6.93 | 61.33 ± 1.15 |
| $NH_4^+$ (mg/L) | 22.50 ± 0.71 [b] | 22.50 ± 2.12 [b] | 26.33 ± 4.5 [b] | 27.33 ± 4.50 [b] | 84.33 ± 7.57 [a] |
| Acetic acid (g/L) | 0.05 ± 0.05 [c] | 0.07 ± 0.03 [c] | 0.29 ± 0.01 [a] | 0.17 ± 0.02 [b] | 0.06 ± 0.01 [c] |

| (c) bottle fermentation | BB1 | RB | BB2 | RB2 | RB3 |
|---|---|---|---|---|---|
| pH | 4.64 ± 0.09 [a] | 4.63 ± 0.10 [a] | 4.30 ± 0.05 [b] | 4.38 ± 0.05 [b] | 4.42 ± 0.03 [b] |
| Alcohol (% *v/v*) | 4.36 ± 0.19 [c] | 4.31 ± 0.14 [c] | 7.17 ± 0.05 [a] | 7.04 ± 0.31 [a] | 5.07 ± 0.07 [b] |
| Maltose (g/L) | 1.30 ± 0.40 [a] | 1.30 ± 0.17 [a] | 0.00 [b] | 0.00 [b] | 0.17 ± 0.15 [b] |
| Glu-Fru (g/L) | 0.01 ± 0.03 | 0.07 ± 0.09 | 0.02 ± 0.01 | 0.05 ± 0.05 | 0.04 ± 0.03 |
| FAN (mg/L) | 115.00 ± 35.38 | 101.33 ± 67.80 | 58.00 ± 5.20 | 55.33 ± 2.89 | 83.00 ± 5.29 |
| $NH_4^+$ (mg/L) | 53.67 ± 28.74 | 56.67 ± 23.86 | 84.33 ± 8.96 | 74.00 ± 3.46 | 61.33 ± 16.74 |
| Acetic acid (g/L) | 0.07 ± 0.03 [b] | 0.11 ± 0.05 [b] | 0.24 ± 0.00 [a] | 0.13 ± 0.02 [b] | 0.075 ± 0.00 [b] |

BB1 = 100% barely malt + *S. cerevisiae* strain Y1; RB1 = rye–barley (55–45%) malt + *S. cerevisiae* strain Y1; BB2 = 100% barely malt + diastatic *S. cerevisiae* strain Y2; RB2 = rye–barley (55–45%) malt + diastatic *S. cerevisiae* strain Y2; RB3 = rye–barley (55–45%) malt + *S. cerevisiae* strain Y3. Glu-Fru = glucose and fructose; FAN = Free amino nitrogen; $NH_4^+$: ammonia, inorganic nitrogen; n.d. = not detected. Data shown are average ± standard deviation of triplicates. Different letters identify significantly different groups ($p < 0.05$) highlighted by one-way ANOVA analysis and Tukey's multiple comparison post hoc test. No letters were written for variables, showing no differences among samples.

### 3.2.1. Sugars and Alcohol

Concerning sugars, fermentable sugars, like maltose, glucose, and fructose, were measured after every step of the beer production process. Before fermentation occurred, the wort density was adjusted to ca. 12.5 °P. At this stage, the maltose detected was similar in all the samples, even if a slight but significant difference was found between samples BB1 and RB3. More pronounced were the differences in glucose and fructose concentration (Table 3). In addition to these differences, there were consistent variations in the sugar concentrations among batches (Table 3a) of the same malt variant. It is known that the mashing step is crucial for the activation of α- and β-amylase and the consequent starch digestion [23–25]. Mashing parameters were previously optimized for each malt grist, and some differences in sugar release can be attributed to differences in the enzymatic activity in each batch. After the first fermentation, maltose was still detectable, and no significant differences among the samples were highlighted. The concentration of glucose and fructose was below 1 g/L in all the samples (Table 3b).

At the end of the second fermentation, only traces of residual glucose and fructose were found throughout the samples, while residual maltose was still measured (Table 3c) in beers fermented with Y1 (1.3 ± 0.40 g/L), and only in traces in those fermented by Y3. Beers produced with Y2 contained no residual maltose. The results indicate that both the Y2 and Y3 yeast strains efficiently utilized the available maltose during the second fermentation, irrespective of the malt variant (barley or rye). These differences are attributable to the ability of yeast to transport maltose into the cell [26].

Concerning ethanol production, after the first and second fermentations, significative differences were observed on the base of the fermenting yeast (Table 3b,c). Beers produced with Y2 had the highest alcohol content in comparison with the other treatments, and no differences were observed among BB2 and RB2, suggesting malt composition had no influence on the ethanol yield for Y2. RB3 beers had a higher alcohol content in comparison with those fermented with Y1. These observations are consistent with the natural features of the inoculated yeast. The most abundant fermentable sugars in brewed wort are maltose

(50–60%), maltotriose (15–20%), and glucose (10–15%) [27]. While glucose is transported passively into the cell by facilitated diffusion, maltose and maltotriose require active transportation into the cell [16], and their uptake generally begins when half of the wort glucose has been consumed. Maltotriose is scarcely utilized by the most common beer yeast strains, which prefer to metabolize glucose as the first choice. The uptake of maltotriose is very slow and can be incomplete, leaving a high content of residual fermentable sugars in the finished product [27]. Despite this, the wort analysis showed no differences in the content of the measurable sugars among the malts; at the end of both the fermentations, BB1 and RB1 beers were characterized by lower alcohol and higher residual maltose in the finished products in comparison with the other beers, suggesting that a less efficient utilization of sugars occurred. This observation agrees with the peculiar feature of the yeast Y1, which is reported by the manufacturer as not being able to use maltotriose and other oligosaccharides. A similar scenario was observed for Y3 beers, while beers produced with Y2 had the lowest sugar content and the highest ethanol concentration. These observations are attributable to the yeast's diastatic nature and are a consequence of the expression of an extracellular glucoamylase that allows Y2 to utilize with more efficiency maltotriose, other oligosaccharides, and starch in addition to maltose, glucose, and fructose, resulting in a super attenuation behavior.

### 3.2.2. Free Amino Nitrogen (FAN) and Ammonium ($NH_4^+$)

Yeast growth involves the uptake of nitrogen, mainly in the form of amino acids, for the synthesis of cellular proteins and other cell compounds. The main nitrogen source in wort is represented by free alpha-amino acids (FAN), small peptides, and ammonium ions ($NH_4^+$) derived from the proteolysis of malt proteins [28]. During beer production, yeasts utilize FAN and ammonium and release a part of them through the autolysis process that occurs at the end of the production process, as also reflected in our study, as shown in Table 3. In all the samples, both FAN and $NH_4^+$ amounts decreased after the first fermentation and increased later at the end of the second fermentation (Table 4). No significative differences in FAN concentration were found in the samples at the same production stage, suggesting that neither the malt variant nor the yeast strain influenced the amount of residual FAN. Regarding the concentration of ammoniacal nitrogen, the amounts detected in RB1 and RB3 wort samples were higher than in the others. After the first fermentation, RB3 beer had a higher amount of ammoniacal nitrogen in comparison with the other beers, suggesting a lower uptake of $NH_4^+$ from the *S. cerevisiae* Y3 strain. No differences were observed at the end of the second fermentation. In general, the amount of FAN and ammonia in all wort samples was sufficient to ensure a good nitrogen supply to the yeast. After the first fermentation, the uptake of nitrogen was observed in all the samples. After bottle fermentation, the release from yeast or from hydrolysis of soluble proteins was detected. The described trend is in line with the results from other studies as reviewed by Hill and Stewart [28].

**Table 4.** Volatile organic compounds in the final beers produced with different *S. cerevisiae* yeast strains after brewing 100% barley or rye–barley (55–45%) malt.

| Class | Compound | BB1 | RB1 | BB2 | RB2 | RB3 |
|---|---|---|---|---|---|---|
| Terpenes | Linalool | 0.056 ± 0.003 [b] | 0.056 ± 0.007 [b] | 0.069 ± 0.010 [ab] | 0.052 ± 0.018 [b] | 0.111 ± 0.011 [a] |
| | Citronellol | 0.009 ± 0.002 [b] | 0.011 ± 0.001 [b] | 0.012 ± 0.003 [b] | 0.011 ± 0.002 [b] | 0.020 ± 0.001 [a] |
| | Humulene | 0.002 ± 0.001 [b] | 0.004 ± 0.001 [b] | 0.001 ± 0.001 [b] | 0.003 ± 0.001 [b] | 0.020 ± 0.003 [a] |
| | Humulene epoxide I | 0.002 ± 0.001 | 0.004 ± 0.002 | 0.005 ± 0.002 | 0.004 ± 0.001 | 0.003 ± 0.001 |
| | Methyl geraniate | 0.007 ± 0.002 [b] | 0.008 ± 0.002 [b] | 0.012 ± 0.002 [b] | 0.010 ± 0.001 [b] | 0.020 ± 0.001 [a] |
| | α-Terpineol | 0.004 ± 0.001 | 0.003 ± 0.001 | 0.003 ± 0.001 | 0.003 ± 0.001 | 0.004 ± 0.001 |
| Sum | | 0.077 ± 0.006 [b] | 0.077 ± 0.012 [b] | 0.093 ± 0.012 [b] | 0.073 ± 0.021 [b] | 0.170 ± 0.010 [a] |

**Table 4.** *Cont.*

| Class | Compound | BB1 | RB1 | BB2 | RB2 | RB3 |
|---|---|---|---|---|---|---|
| Alcohols | 2-Methyl- 1-propanol | 0.326 ± 0.075 [b] | 0.314 ± 0.030 [b] | 0.280 ± 0.018 [b] | 0.280 ± 0.053 [b] | 0.545 ± 0.036 [a] |
| | 3-Methyl- 1-butanol | 3.200 ± 0.681 [ab] | 3.085 ± 0.333 [ab] | 4.433 ± 1.275 [ab] | 5.142 ± 1.080 [a] | 2.384 ± 0.07 [b] |
| | 1-Hexanol | 0.017 ± 0.002 [a] | 0.018 ± 0.002 [a] | 0.013 ± 0.002 [b] | 0.016 ± 0.004 [ab] | 0.021 ± 0.004 [a] |
| | 2,3-Butanediol | 0.013 ± 0.005 | 0.013 ± 0.002 | 0.013 ± 0.006 | 0.009 ± 0.002 | 0.016 ± 0.005 |
| | 2-Decanol | 0.012 ± 0.001 [bc] | 0.015 ± 0.002 [bc] | 0.019 ± 0.005 [ab] | 0.012 ± 0.004 [bc] | 0.029 ± 0.005 [a] |
| | Phenylethyl alcohol | 5.075 ± 0.922 | 4.827 ± 0.573 | 5.232 ± 0.459 | 4.641 ± 1.024 | 3.564 ± 0.143 |
| | 2-Methoxy-4-vinylphenol | 0.004 ± 0.001 [b] | 0.006 ± 0.001 [b] | 0.017 ± 0.009 [b] | 0.047 ± 0.013 [a] | 0.006 ± 0.002 [b] |
| Sum | | 8.647 ± 1.674 | 8.277 ± 0.923 | 10.008 ± 1.423 | 10.148 ± 2.173 | 6.565 ± 0.221 |
| Esters | Ethyl acetate | 0.627 ± 0.165 [b] | 0.688 ± 0.123 [b] | 1.235 ± 0.412 [a] | 1.222 ± 0.211 [a] | 0.885 ± 0.045 [ab] |
| | 1-Butanol, 3-methyl-, acetate | 0.674 ± 0.371 [b] | 1.363 ± 0.787 [b] | 3.295 ± 1.873 [a] | 4.520 ± 0.762 [a] | 0.813 ± 0.082 [b] |
| | Hexanoic acid, ethyl ester | 0.449 ± 0.139 [b] | 0.539 ± 0.140 [b] | 0.834 ± 0.177 [a] | 0.854 ± 0.168 [a] | 0.767 ± 0.098 [a] |
| | Heptanoic acid, ethyl ester | 0.042 ± 0.015 [b] | 0.044 ± 0.011 [b] | 0.057 ± 0.011 [b] | 0.066 ± 0.012 [ab] | 0.101 ± 0.012 [a] |
| | Octanoic acid, ethyl ester | 8.612 ± 2.515 | 8.610 ± 2.135 | 13.137 ± 2.204 | 12.445 ± 2.322 | 8.776 ± 0.137 |
| | Isopentyl hexanoate | 0.033 ± 0.016 [b] | 0.032 ± 0.013 [b] | 0.079 ± 0.027 [ab] | 0.105 ± 0.030 [a] | 0.047 ± 0.003 [b] |
| | Nonanoic acid, ethyl ester | 0.269 ± 0.077 | 0.353 ± 0.111 | 0.535 ± 0.156 | 0.400 ± 0.117 | 0.384 ± 0.049 |
| | Decanoic acid, ethyl ester | 4.679 ± 1.597 | 4.315 ± 2.021 | 6.191 ± 1.734 | 4.358 ± 0.576 | 2.296 ± 0.103 |
| | Ethyl (Z)-4-decenoate | 0.134 ± 0.029 | 0.274 ± 0.157 | 0.483 ± 0.230 | 0.401 ± 175 | 0.077 ± 0.017 |
| | 2-phenylethyl acetate | 0.714 ± 0.332 [b] | 1.175 ± 0.647 [a] | 1.621 ± 0.318 [a] | 1.497 ± 0.402 [a] | 0.439 ± 0.060 [b] |
| | Ethyl hexadecanoate | 0.027 ± 0.006 | 0.022 ± 0.006 | 0.031 ± 0.006 | 0.025 ± 0.004 | 0.015 ± 0.001 |
| | Butyl hexadecanoate | 0.005 ± 0.002 | 0.005 ± 0.002 | 0.007 ± 0.003 | 0.007 ± 0.003 | 0.010 ± 0.002 |
| Sum | | 16.256 ± 3.725 [b] | 17.427 ± 5.939 [b] | 27.513 ± 4.959 [a] | 25.900 ± 4.411 [a] | 14.600 ± 0.283 [b] |
| Acids | 3-Decenoic acid | 0.006 ± 0.002 [b] | 0.005 ± 0.002 [b] | 0.008 ± 0.002 [b] | 0.007 ± 0.001 [b] | 0.015 ± 0.002 [a] |
| | Heptanoic acid | 0.284 ± 0.036 [ab] | 0.317 ± 0.051 [ab] | 0.477 ± 0.105 [a] | 0.447 ± 0.124 [a] | 0.166 ± 0.010 [b] |
| | Octanoic acid | 1.208 ± 0.009 [ab] | 1.385 ± 0.305 [b] | 1.705 ± 0.227 [a] | 1.333 ± 0.336 [b] | 0.412 ± 0.039 [b] |
| | Nonanoic Acid | 0.012 ± 0.001 | 0.018 ± 0.003 | 0.014 ± 002 | 0.010 ± 0.002 | 0.005 ± 0.001 |
| | n-Decanoic acid | 0.207 ± 0.017 [a] | 0.164 ± 0.072 [ab] | 0.172 ± 0.057 [ab] | 0.105 ± 0.017 [ab] | 0.030 ± 0.001 [b] |
| Sum | | 1.717 ± 0.011 [b] | 1.889 ± 0.423 [ab] | 2.376 ± 0.282 [a] | 1.901 ± 0.479 [ab] | 0.629 ± 0.051 [c] |
| Others | 2-Decanone | 0.003 ± 0.001 | 0.007 ± 0.002 | 0.004 ± 0.001 | 0.003 ± 0.001 | 0.002 ± 0.001 |
| | γ-Butylbutyrolactone | 0.029 ± 0.003 | 0.029 ± 0.010 | 0.027 ± 0.003 | 0.033 ± 0.007 | 0.029 ± 0.002 |
| | γ-Decalactone | 0.004 ± 0.001 | 0.004 ± 0.001 | 0.004 ± 0.001 | 0.003 ± 0.001 | 0.002 ± 0.001 |
| | 2-Acetylpyrrole | 0.006 ± 0.001 [ab] | 0.004 ± 0.001 [b] | 0.009 ± 0.001 [a] | 0.003 ± 0.001 [b] | 0.004 ± 0.001 [b] |
| Sum | | 0.042 ± 0.005 | 0.044 ± 0.012 | 0.044 ± 0.003 | 0.042 ± 0.008 | 0.037 ± 0.002 |

Volatile organic compounds are expressed as mg/L 1-heptanol. Data shown are average ± standard deviation of triplicates. Different letters identify significantly different groups ($p < 0.05$) of beer at the end of fermentation after one-way ANOVA analysis and Tukey's multiple comparison post hoc test. No letters were written for variables, showing no differences among samples.

### 3.2.3. Acetic Acid

After the first fermentation, beers obtained with Y2 showed a significantly higher amount of acetic acid in comparison with the other samples, as shown in Table 3b, with BB2 being characterized by the highest acetic acid concentration (0.29 ± 0.01 g/L). At the end of the maturation process (Table 3c), the concentration of acetic acid slightly increased in all the beers apart from BB2 and RB2. Although similar amounts of acetic acid have been detected in all the samples, BB2 has shown the significantly highest concentration. Acetic acid represents a by-product of alcoholic fermentation and is often correlated to the yeast strain fermenting metabolism. The amount of acetic acid in beer is highly associated with the off-flavor, which makes the beer unattractive [29]. Data in the literature defined the odor threshold for acetic acid in water at 0.18 g/L [30]. According to this observation, in our study, only BB2 had a concentration that exceeded the odor threshold in water; however, no off-flavor was associated with this sample in the sensory evaluation.

### 3.3. Volatile Compounds

In this study, 34 volatile compounds were identified within the beers, each contributing to the aromatic tapestry of these brews at the end of the production process (Table 4). In terms of abundance, the prevailing aromatic profile of these beverages was dominated by the presence of esters, alcohols, and acids and, in minor part, by terpenes and other compounds. A two-way analysis, reported in Table 4, revealed that yeast rather than malt greatly contributes to shaping the aroma profile of beers. The concentration of thirteen volatile compounds was influenced by the yeast, one by the malt, and one by the interaction of malt and yeast, suggesting that the choice of the yeast is crucial to confer the desired flavor characteristic to the product. In particular, the yeast had a big influence on the total concentration of terpenes and acids as well as for some esters and alcohols, as better described in the following sections. The amount of 2-acetylpyrrole was influenced by the malt utilized for beer production, being higher in barley beers than in rye–barley beers, while the nonanoic acid was influenced by both yeast and the interactions between yeast and malt.

#### 3.3.1. Terpenes

Terpenes are important flavors in beer. Generally deriving from hop, these molecules are bio-transformed by yeasts during the fermentation process [31]. The analysis identified six terpenes, with linalool, associated with aniseed and lemon flavor [32], being the major compound of this class. Because of its pleasant aroma and low odor threshold (27 ng/L), linalool is considered one of the most desirable oil compounds in beer [33]. Beers produced with Y3 yeast showed the highest concentration of linalool, methyl geraniate, citronellol, and humulene in comparison with the other samples. All these terpenes are associated with herbal and flowery notes, conferring characteristic fragrance to the beers.

#### 3.3.2. Alcohols

During fermentation, volatile alcohols are formed as by-products of amino acid synthesis from pyruvate or through amino acid catabolism, and their composition and concentration are influenced by the wort and yeast metabolism [34]. In the present study, seven volatile alcohols were detected besides ethanol (see Section 3.2.1). ANOVA analysis highlighted significant differences for five compounds: 2-methyl-1-propanol, 2-methoxy-4-vinylphenol, 2-decanol, 1-hexanol, and 3-methyl-1-butanol, as shown in Table 4.

The most abundant alcohols were the phenylethyl alcohol, characterized by a rose-honey aroma, and the 3-methyl-1-butanol, associated with a malty aroma [35]. These alcohols are commonly found in beer, and, depending on concentration, they may have positive and negative impacts on the flavor, conferring fruity, floral notes, or a pungent smell. In our study, fusel alcohols were present at concentrations lower than 300 mg/L, indicated as the threshold for the development of unpleasant flavors [34].

#### 3.3.3. Esters

Esters have an important role in the aroma of beers and fermented beverages. Most are volatile compounds produced during the fermentation process as a result of yeast metabolism [35]. As ester production is closely linked to yeast metabolism, the choice of fermenting yeast is fundamental for the development of the beer aroma. When present in moderate quantities, the ester synergy contributes to adding a pleasant, full-bodied character to the beer and shaping a peculiar product. On the other hand, an excessive production of esters may lead to an undesirable fruity aroma. Two main ester groups are present in fermented beverages: acetate esters and ethyl esters [35]. As reported in Table 4, at the end of the production process, octanoic, decanoic, and hexanoic ethyl esters and the ethyl-, 3-methyl-1-butanol-, and 2-phenylethyl-acetate were produced in higher amounts. These results agree with data reported in the literature describing the above-mentioned metabolites as the main representative esters in beers [32,36].

Among the samples, significant differences were detected in ester concentration on the base of the fermentative yeast (Table 4). Both barley and rye–barley beers fermented by Y2 had the highest content in esters compared with the other samples. All beers were dominated by octanoic acid ethyl ester, often associated with a sweaty, fatty flavor [37]. The ANOVA analysis allowed us to find significative differences for seven esters. Beers fermented with Y2 contained significantly higher concentrations of ethyl acetate, isoamyl acetate (3-methyl-1-butanol acetate), hexanoic acid ethyl ester (comparable with RB3 beers), isopentyl hexanoate, and acetic acid 2-phenylethyl ester. Isoamyl acetate is characterized by a banana flavor, while the hexanoic acid ethyl ester confers a fatty note to the beer aroma [37]. Yeast Y3 was found to induce a higher production of heptanoic acid, ethyl ester, associated with a fruity candy flavor [38], in comparison with the other yeasts.

### 3.3.4. Acids

Volatile acids are quantitatively minor constituents in beer. They play an important role in fermented beverages, conferring acidity, participating in the formation of volatile compounds that enrich the aromatic bouquet of the products, and stabilizing the foam. Besides acetic acid, in our study, fatty acids C7, C8, and C10 acids were detected, with octanoic acid being the compound with the highest concentration, followed by heptanoic and decanoic acid. When present in excess, they are responsible for rancid or goat flavor characteristics [39]. With the exception of nonanoic acid, all the compounds identified in this study showed significant differences in concentration among the samples. Beers fermented by Y3 were characterized by the lowest amounts of volatile acids.

### 3.3.5. Other Compounds

Other compounds were identified: two lactones, one cheton, and one pyrrole, as described in Table 4.

Lactones are organic compounds with a distinctive peach- and coconut-like aroma, produced during fermentation by the hydroxylation and oxidation of free fatty acids. The concentration of lactones is positively correlated with the duration of the fermentation. In our samples, we detected γ-decalactone, which is synthetized from oleic acid [40], and γ-butylbutyrolactone with no differences among the samples.

In addition to lactones, one cheton was detected, namely 2-decanone, described as an indicator of enzymatic decarboxylation of fatty acids [41].

Finally, 2-acetylpyrrole, formed through Maillard reactions during the malting and brewing processes [42], was detected.

### 3.4. Sensory Evaluation and VOC Principal Component Analysis (PCA)

To better understand the effect of the applied treatments (malt and yeast), the sensory evaluation results were combined with the VOC profile of the beers in the PCA analysis. The individual compounds and sensory parameters were used as variables to explain the differences among beer samples, as shown in Figure 2. The application of the SPME technique enabled the discrimination between yeast strains Y1, Y2, and Y3 in terms of their aroma compound production, as also highlighted by the PCA.

The first two components explained 49.3% of the variance (PC1 27.5%, PC2 21.8%) and allowed a separation of the samples in the plane (Figure 2). In particular, the separation was more evident along the PC1. Here, samples fermented with Y3 were grouped on the right side, while beers obtained with Y1 and Y2 were placed on the left side. Beers fermented with Y1 and Y2 partially overlapped, but a separation on the PC2 was visible. All the beers fermented with Y1 were placed on the upper part of the graphic, while beers obtained with Y2, except for two samples, were placed on the lower part of the plane.

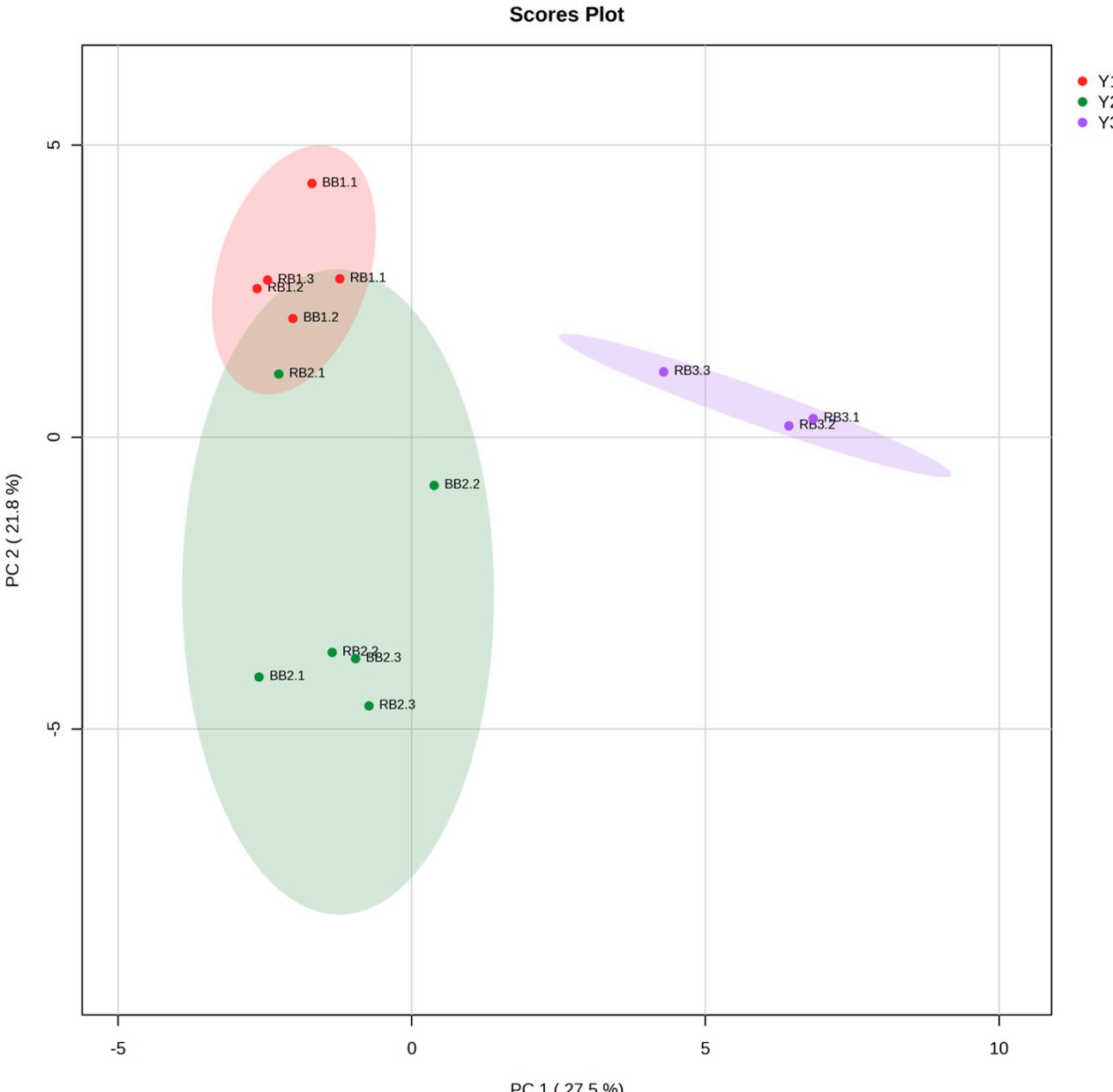

**Figure 2.** Score plot of the principal component analysis of volatile compounds and sensory attributes of beers. Sample names and variable names are indicated as follows: (Y1) *S. cerevisiae* London ESB., (Y2) *S. cerevisiae* var. *diastaticus* Belle Saison, (Y3) *S. cerevisiae* BRY-97 West Coast Ale, (BB) 100% barley beers, (RB) 55–45% rye–barley beers.

Y3 fermented beers were characterized by a higher amount of terpenes (linalool, methyl geraniate, humulene, citronellol, and terpineol) of some esters and of the alcohols 2-methyl-1-propanol, decanol, and hexanol, 2,3-butanediol (Figure 3). The sensorial components overall impression, bitterness, and perlage positively correlated with the VOC Y3 beer discriminants.

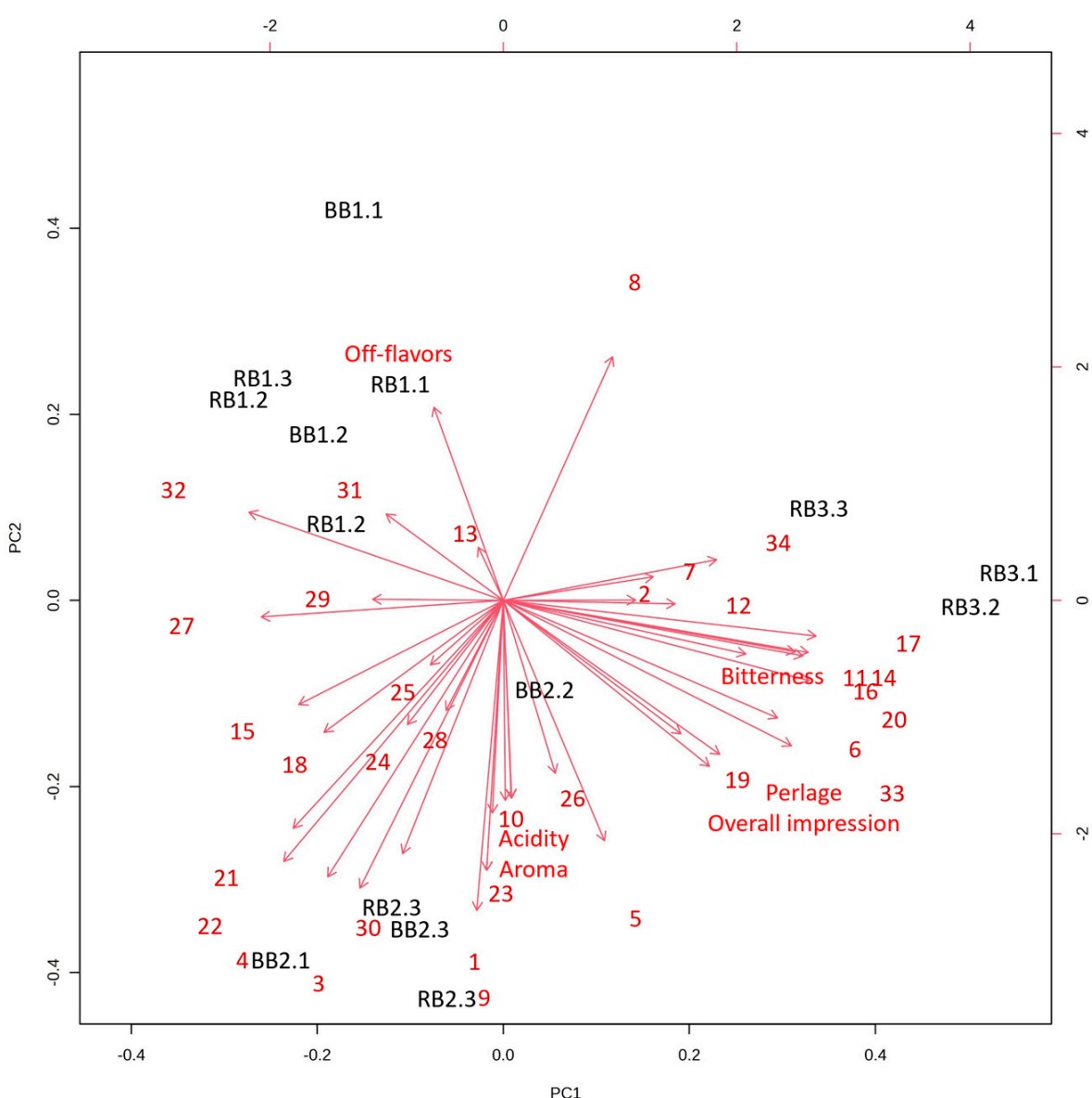

**Figure 3.** Biplot of the principal component analysis of volatile compounds and sensory attributes of beers. Sample names (black) and variable names (red) are indicated as follows: BB1 = 100% barely malt + *S. cerevisiae* strain Y1; RB1 = rye–barley (55–45%) malt + *S. cerevisiae* strain Y1; BB2 = 100% barely malt + diastatic *S. cerevisiae* strain Y2; RB2 = rye–barley (55–45%) malt + diastatic *S. cerevisiae* strain Y2; RB3 = rye–barley (55–45%) malt + *S. cerevisiae* strain Y3. (Y1) *S.cerevisiae* London ESB., (Y2) *S. cerevisiae* var. *diastaticus* Belle Saison, (Y3) *S. cerevisiae* BRY-97 West Coast Ale. Compounds are indicated as numbers in the biplot: (1) Ethyl acetate; (2) 2-methyl-1-propanol; (3) 3-methyl-1-butanol, acetate; (4) 3-methyl-1-butanol; (5) Hexanoic acid ethyl ester; (6) Heptanoic acid ethyl ester; (7) 1-Hexanol; (8) Linalool; (9) Octanoic acid ethyl ester; (10) Isopentyl hexanoate; (11) Nonanoic acid ethyl ester; (12) 2,3-Butanediol; (13) 2-Decanone; (14) 2-Decanol; (15) Decanoic acid ethyl ester; (16) 3-Decenoic acid; (17) Humulene; (18) Ethyl (Z)-4-decenoate; (19) Methyl geraniate; (20) Citronellol; (21) 2-phenylethyl acetate; (22) Heptanoic acid; (23) Phenylethyl alcohol; (24) 2-Acetylpyrrole; (25) Humulene epoxide I; (26) γ-Butylbutyrolactone; (27) Octanoic acid; (28) γ-Decalactone; (29) Nonanoic acid; (30) 2-Methoxy-4-vinylphenol; (31) Hexadecanoic acid, ethyl ester; (32) n-Decanoic acid; (33) Hexanoic acid, butyl ester; (34) α-Terpineol. Overall impression, perlage, off-flavor, aroma, and acidity (the latter two are superimposed) represent the evaluated sensory attributes.

Y1 beers contained the major quantities of the acids decanoic, hexanoic, and octanoic acid and obtained the highest off-flavor scores in comparison with the other beers (Table 4, and Figure 3) Y2 beers had a higher concentration of esters and phenyl ethyl alcohol and obtained the highest evaluation for acidity and aroma (Figure 3). Altogether, our data suggest that Y2 could be further evaluated for use in brewing. The use of *S. cerevisiae* var. *diastaticus* is often limited by the natural feature of the yeast strains, often POF-positive, therefore forming phenolic off-flavors through the action of yeast decarboxylases [43]. In our study, the POF-negative Y2 strain received a positive overall impression, and no off-flavor notes were highlighted by the panelists, suggesting Y2 could find application in the development of novel beers with unique aromas and organoleptic properties.

### 3.5. Influence of Malt and Yeast on Beer Parameters

All the data collected for the final products, including the physicochemical parameters, the volatile compounds, and the sensory perception, underwent a two-way ANOVA analysis. Twenty parameters were significantly influenced by malt, yeast, or their interactions (Table 5).

**Table 5.** F-values and significant differences of physicochemical, volatile, and sensorial variables in beers in relation to malt and yeast obtained by a two-way ANOVA and Bonferroni correction.

| | Yeast | Malt | YxM |
|---|---|---|---|
| Physicochemical parameters | | | |
| Ethyl alcohol | 378.56 *** | 0.803 | 0.166 |
| Brix | 91.43 *** | 0.242 | 0.544 |
| pH | 27.60 ** | 0.73 | 1.207 |
| Acetic acid | 23.82 ** | 4.201 | 26.945 * |
| Volatile compounds | | | |
| Humulene | 77.00 ** | 0.804 | $1.65 \times 10^{-4}$ |
| Maltose | 64.89 ** | 0 | 0 |
| 2-Methyl-1 propanol | 38.71 ** | 0.179 | 0.199 |
| Terpene Sum | 35.46 ** | 1.407 | 1.103 |
| Methyl geraniate | 32.46 ** | 1.621 | 0.409 |
| Linalool | 25.47 ** | 2.721 | 1.01 |
| 3-Decenoic acid | 23.75 ** | 1.437 | 0.001 |
| 2-Methoxy-4-vinylphenol | 23.01 ** | 9.336 | 13.304 |
| Nonanoic Acid | 21.46 * | 0.289 | 21.379 * |
| Citronellol | 20.04 * | 0.002 | 0.458 |
| Octanoic acid | 18.75 * | 0.548 | 4.359 |
| Acid Sum | 18.10 * | 0.675 | 3.248 |
| 2-Decanol | 16.78 * | 1.981 | 3.206 |
| Heptanoic acid, ethyl ester | 15.33 * | 0.049 | 0.886 |
| Hexanoic acid, butyl ester | 14.69 * | 0.05 | 0.732 |
| 2-Acetylpyrrole | 2.07 | 44.644 ** | 9.691 |

YxM: Yeast and malt interaction. *** $p < 0.001$; ** $p < 0.01$; * $p < 0.05$.

Yeast significantly influenced 17 variables, suggesting that yeast rather than malt greatly contributes to shaping the beer characteristics. In particular, the yeast was able to influence the alcohol production and the aromatic characteristics of beers by modulating the total concentration of terpenes and acids as well as of some esters and alcohols, as better described in Section 3. Differently, the amount of 2-acetylpyrrole was influenced by the malt utilized for beer production, being higher in barley beers than in rye–barley beers.

The concentration of acetic acid and nonanoic acid appeared to be the interplay result of the yeast metabolism and the malt composition. Because little information is available

about the complex influence of yeast and malt participating in the beer fermentation process [44,45], further investigation should be carried out to better understand the metabolic basis underneath.

## 4. Conclusions

The outcomes of this investigation present discernible differences, underscoring the influence of yeast selection on the resultant aromatic bouquet of the beers. The evidence of a compelling case for the superior aromatic potential of the yeast strain Y2 was highlighted. The heightened synthesis of a diverse array of aromatic compounds culminates in a beer characterized by an enhanced aromatic profile. The ability of Y2 to produce a greater abundance of aromas manifests in the sensory experience, yielding a beer that is distinctly more aromatic and thereby likely to captivate the discerning palates of consumers. In essence, yeast selection becomes an indispensable facet in the pursuit of crafting beers that transcend their gustatory appeal, enveloping enthusiasts in a multi-sensory delight, and deserve further investigation in the upscale process.

**Supplementary Materials:** The following supporting information can be downloaded at https://www.mdpi.com/article/10.3390/beverages9040093/s1. Figure S1: Total ion chromatogram of rye–barley beers used as quality control.

**Author Contributions:** Conceptualization, L.C.; Methodology, G.M.R., A.R.G., P.H., P.R. and L.C.; Software, N.T.; Validation, N.T., G.M.R., A.R.G. and P.R.; Formal analysis, N.T., G.M.R., A.R.G. and P.H.; Investigation, A.R.G. and P.H.; Resources, P.R. and L.C.; Data curation, G.M.R.; Writing—original draft, A.R.G.; Writing—review & editing, N.T., G.M.R., P.R. and L.C.; Visualization, N.T.; Project administration, L.C.; Funding acquisition, P.R. and L.C. All authors have read and agreed to the published version of the manuscript.

**Funding:** This research was funded by the Autonomous Province of Bozen-Bolzano (Action plan for mountain agriculture, Decision n. 1016, 1 September 2015).

**Data Availability Statement:** The data used to support the findings of this study can be made available by the corresponding author upon request.

**Acknowledgments:** All of the authors would like to acknowledge the funding agency Provincia autonoma di Bolzano-Alto Adige- ITALY—Ripartizione Innovazione, Ricerca e Università, for funding the project titled "Brewing in Circle: design and implementation of South Tyrolean craft seasonal beer process recovering functional by-product (CirBeer)" CUP H34I19001090003. The authors thank the Department of Innovation, Research, University, and Museums of the Autonomous Province of Bozen/Bolzano for covering the open access publication costs.

**Conflicts of Interest:** The authors declare no conflict of interest. The funders had no role in the design of the study; in the collection, analyses, or interpretation of data; in the writing of the manuscript; or in the decision to publish the results.

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
