# Peer review of "The Impact of Rye and Barley Malt and Different Strains of Saccharomyces cerevisiae on Beer Volatilome"

_beverages, doi:10.3390/beverages9040093_

Round 1

Reviewer 1 Report

Comments and Suggestions for Authors

The topic is very interesting but it requires greater study of the effects of raw materials and yeasts.

It should be appropriate to modify the manuscript according to the following suggestions:

-Lines 26-30: concerning economic data, authors are requested to report more recent data. 

-Lines 30-32: authors are requested to report data concerning Italy and Europe in general (not only Germany and UK).

-Table 2: the experimental design is not very clear. Please, specify the samples and the corresponding ingredients. I have the sensation that a sample (BB3) lacks. 

-A Two-Way ANOVA could better describe the effects of malt and yeast.

-The experimental results must be discussed more in depth.

Author Response

To REVIEWER 1 Reply to Comments and Suggestions for Authors

Thank you very much to the Reviewer for the precious suggestions and comments

Q1: Lines 26-30: concerning economic data, authors are requested to report more recent data.

R1 The introduction has been updated with more recent data, as suggested by the reviewer.

Q2: Lines 30-32: authors are requested to report data concerning Italy and Europe in general (not only Germany and UK).

R2: The Introduction has been enriched with more and recent information on the Italian and European reality of craft breweries.

Q3: Table 2: the experimental design is not very clear. Please, specify the samples and the corresponding ingredients. I have the sensation that a sample (BB3) lacks.

R3: Table 2 has been modified to increase its clarity. The yeast strain 3 was used only for RB malt as specifically selected for rye. The intent was to have a parameter to compare the rye-barley malt fermentation with several yeasts. This description was added in the material and method section: lines 100-104.

Q4: A Two-Way ANOVA could better describe the effects of malt and yeast.

R4: A two-way ANOVA analysis was performed to better discriminate the effect of yeast and malt as suggested and data are reported in the new table 4.

Q5: The experimental results must be discussed more in depth.

R5: According to the reviewer comment, we performed a two-way ANOVA analysis and the results, added in table 4, were reported in the text, to discuss more in depth about the effect of yeast and malt on the volatilome of the beers: section 3.3.

Reviewer 2 Report

Comments and Suggestions for Authors

In this manuscript “Impact of rye and barley malt and different strains of Saccharomyces cerevisiae on beer volatilome”. The research discusses the exploration of the behavior of three different strains of Saccharomyces cerevisiae yeast in the fermentation of ale beer, particularly when using a high proportion of rye malt compared to pure barley malt. Craft breweries are interested in creating beers with locally sourced raw materials and distinctive flavor profiles to meet consumer demands.

In this study, 34 volatile organic compounds were identified in the beer. The research findings suggest that the choice of yeast strain has a more significant impact on the beer's volatile profile than the type of malt used. However, there are still some problems in the manuscript. The comments and problems are as follows:

1. Why the CO2 content at the beginning of fermentation(0h) was not measured.

2. There is a problem with the format of Table 3. There is also a naming problem in Table 3. Table 5 and Table 6 appeared in the text, which was strange.

3. In PCA analysis, the number of samples per group is inconsistent and small and does not correspond to the number of 37 mentioned in part 2.5.

4. The experimental grouping should add the BB3 group if we follow the analogy of other groups, why not?

5. In the first paragraph of the Introduction, when it comes to the production of beer, the data is from 2017, so it is suggested to replace it with a more updated one.

Author Response

Inizio modulo

To REVIEWER 2 Reply to Comments and Suggestions for Authors

Thank you very much to the Reviewer for the precious suggestions and comments

Q1: Why the COcontent at the beginning of fermentation(0h) was not measured.

R1: We considered the day 0 as the day of yeast inoculation. At this stage there is no CO2 production, as CO2 is a product of the fermentation process. We added a point in the figure 1 in order to show also Day 0.

Q2: There is a problem with the format of Table 3. There is also a naming problem in Table 3. Table 5 and Table 6 appeared in the text, which was strange.

R2: The table 3 has been revised and corrected and table 5 and 6 have been removed throughout the text.

Q3: In PCA analysis, the number of samples per group is inconsistent and small and does not correspond to the number of 37 mentioned in part 2.5.

R3: The variables considered for PCA analysis were represented by volatile compounds, and sensory attributes obtained from the analysis of 15 beer samples. The points in PCA score plot represent the samples, while the arrows in the bi-plot are the variables.  37 were the judges participating to the tasting (as defined in section 2.5) and their evaluations have been processed to obtain a mean and standard deviation value for each sensory variable (5 attributes and presence of off-flavors) for each beer sample. In order to help to clarify this fact we have made some changes in the figure 2 and 3.

Q4: The experimental grouping should add the BB3 group if we follow the analogy of other groups, why not?

R4: The yeast strain 3 was used only for RB malt as specifically selected for rye. The intent was to have a parameter to compare the rye-barley malt fermentation with several yeasts. This description was added in the material and method section: lines 100-104.

Q5: In the first paragraph of the Introduction, when it comes to the production of beer, the data is from 2017, so it is suggested to replace it with a more updated one.

R5: The Introduction has been updated with more recent data.

Reviewer 3 Report

Comments and Suggestions for Authors

The manuscript describes the Impact of rye and barley malt and different strains of Saccharo-myces cerevisiae on beer volatilome. The topic is relevant to the aim and scope of the beverages. The manuscript is well written and easy to follow. Some clarifications in the texts are needed. Overall, this manuscript meets the standard for acceptance after addressing the below comments:

1.  Please describe in the manuscript the reason why the 55%-45% was focused on specifically in experimental set up.

2. Please describe in the manuscript the reason why the DVB-CAR-PDMS is used. Furthermore, what is the role of each component, i.e. DVB’s role, CAR’s role, and PDMS’s role.

3. In line 153, “coating 50/30 μm” has been found. What does 50/30 mean? And what is the material of 50/30 μm?

4. In Figure 1, why are not the CO2 productions in BB2 and RB2 saturated although those in others are saturated?

5. The title numbers are confusing. After “3.2.1. Sugars and Alcohol”, “3.1.1. Free amino nitrogen and Ammonium” is shown up. What is the criteria of the numbering?

6.   Figure 2 and 3 are too small to recognize the letters of the figures.

Author Response

Inizio modulo

To REVIEWER 3 Reply to Comments and Suggestions for Authors

Thank you very much to the Reviewer for the precious suggestions and comments

 Q1: Please describe in the manuscript the reason why the 55%-45% was focused on specifically in experimental set up.

R1: The use of rye malt in brewing represents a challenge for brewers due to its specific features such as the the lack of a hull, dense packing in steep, and high wort viscosity that make the brewing process complicated. Here we used a malt composed of 55%-45% rye-barley, selected as the result of past trials (data not shown), to evaluate the influence of yeasts on the physicochemical and sensory profile of a beer brewed with a malt composed in prevalence of rye. A sentence has been added to the Introduction.

Q2:  Please describe in the manuscript the reason why the DVB-CAR-PDMS is used. Furthermore, what is the role of each component, i.e. DVB’s role, CAR’s role, and PDMS’s role.

R2: Accordingly to reviewer´s suggestion we added in the paper the reason why we used the DVB/CAR/PDMS fiber including the role of each component, section 2.4.

Q3.     In line 153, “coating 50/30 μm” has been found. What does 50/30 mean? And what is the material of 50/30 μm?

R3: coating 50/30 µm” means the thickness of the coating: 50 μm for the DVB layer and 30 μm for the CAR/PDMS layer, usually these parameters are not explained in scientific papers, since they are (standard) product specifications of the fiber. Section 2.4.

Q4:      In Figure 1, why are not the CO2 productions in BB2 and RB2 saturated although those in others are saturated?

R4 After the last time point reported, the loss of weight due to CO2 production was not measured, because this was established as end point for all the beers.

Q4:    The title numbers are confusing. After “3.2.1. Sugars and Alcohol”, “3.1.1. Free amino nitrogen and Ammonium” is shown up. What is the criteria of the numbering?

R5: We are sorry for the refuse. The title numbering has been corrected.

Q6:      Figure 2 and 3 are too small to recognize the letters of the figures.

R6: Figure 2 is generated by the tool utilized for the PCA analysis and unfortunately it is no possible to modify the character size. We improved the figure by changing color gradient and substituting the cross indicating the samples with a full dot, making the points more visible. Figure 3 has been improved and letters are now bigger.

Round 2

Reviewer 1 Report

Comments and Suggestions for Authors

Line 30: too many points…

Line 32: please, put a blank between ‘2027’ and ‘[3]’

In the first review, I ask for a Two-Way ANOVA to better describe the effects of malt and yeast. It must be applied to all parameters, not only to volatile compounds. 

The second column of Table 4 should be better explained. Readers can only guess that ‘Y’ stays for yeast, ‘M’ stays for malt, and ‘YM’ stays for yeast*malt. Furthermore, MYM seems to be put together. 

Author Response

To REVIEWER 1 Reply to Comments and Suggestions for Authors

Thank you very much to the Reviewer for the precious suggestions and comments.

Q1: Line 30: too many points…

R1: Line 30 has been revised and corrected.

Q2: Line 32: please, put a blank between ‘2027’ and ‘[3]’

R2: The space has been added.

Q3: In the first review, I ask for a Two-Way ANOVA to better describe the effects of malt and yeast. It must be applied to all parameters, not only to volatile compounds.

R3: Thanks for the valuable comment. The analysis has been performed including all the parameters and now the results are reported as Table 5 and a section 3.5 has been added to the text.

Q4: The second column of Table 4 should be better explained. Readers can only guess that ‘Y’ stays for yeast, ‘M’ stays for malt, and ‘YM’ stays for yeast*malt. Furthermore, MYM seems to be put together.

R4: Table 4 has been modified and a new Table with the results of the two-way ANOVA analysis has been added (Table 5) also taking into account the valuable comment for a better explanation of the labels.

Reviewer 3 Report

Comments and Suggestions for Authors

All has been addressed except Q1. I have not been able to find the description for Q1. Furthermore, the reference for the past trial should have been added in the manuscript. If I was not able to find the reference, please highlight the reference number.

Author Response

To REVIEWER 3 Reply to Comments and Suggestions for Authors

Thank you very much to the Reviewer for the precious suggestions and comments.

Q1: All has been addressed except Q1. I have not been able to find the description for Q1. Furthermore, the reference for the past trial should have been added in the manuscript. If I was not able to find the reference, please highlight the reference number.

R1: We are sorry for forgetting to add a description for question 1 in the first revision.

Rye is a crop uncommonly tolerant to cool climate and less-fertile soil conditions, growing often where wheat cannot be grown, such in the mountain regions. We wanted to explore the possibility to produce a beer made with mountain local ingredient.

The use of rye in brewing is challenging, therefore we perform preliminary trial with rye up to 65% to test for the maximum amount of rye malt to develop a recipe  with a malt composed in prevalence of rye. We didn´t publish these data till now, thus we cannot add a reference.

The description of this fact has been added in experimental set up section (lines 89-94).
